# Blinatumomab in Pediatric Acute Lymphoblastic Leukemia—From Salvage to First Line Therapy (A Systematic Review)

**DOI:** 10.3390/jcm10122544

**Published:** 2021-06-08

**Authors:** Manon Queudeville, Martin Ebinger

**Affiliations:** Department I–General Pediatrics, Hematology/Oncology, Children’s Hospital, University Hospital Tübingen, 72076 Tübingen, Germany; martin.ebinger@med.uni-tuebingen.de

**Keywords:** acute lymphoblastic leukemia, immunotherapy, bispecific T-cell engager (BiTE)

## Abstract

Acute lymphoblastic leukemia is by far the most common malignancy in children, and new immunotherapeutic approaches will clearly change the way we treat our patients in future years. Blinatumomab is a bispecific T-cell-engaging antibody indicated for the treatment of relapsed/refractory acute lymphoblastic leukemia (R/R-ALL). The use of blinatumomab in R/R ALL has shown promising effects, especially as a bridging tool to hematopoietic stem cell transplantation. For heavily pretreated patients, the response to one or two cycles of blinatumomab ranges from 34% to 66%. Two randomized controlled trials have very recently demonstrated an improved reduction in minimal residual disease as well as an increased survival for patients treated with blinatumomab compared to standard consolidation treatment in first relapse. Current trials using blinatumomab frontline for high-risk patients or as a consolidation treatment post-transplant will show whether efficacy is even higher in less heavily pretreated patients. Due to the distinct pattern of adverse events compared to high-dose conventional chemotherapy, blinatumomab could play an important role for patients with a risk for severe chemotherapy-associated toxicities. This systematic review discusses all published results for blinatumomab in children as well as all ongoing clinical trials.

## 1. Introduction

Blinatumomab is a bispecific T-cell engaging (BiTE) antibody linking the targeting regions of two antibodies directed against CD19 and CD3. CD19 is expressed by the precursor-B-ALL cells, and CD3 is the constant part of the T-cell receptor (TCR) complex that mediates T-cell receptor signaling. Blinatumomab, therefore, leads to a very close linkage between malignant B cells and T cells, a cytolytic synapse forming in the close contact zone [1]. Multiple, bivalent binding leads to a strong stimulus of the engaged T cell which is independent of the TCR specificity and of MHC class I antigen presentation or other costimulatory factors [2,3]. The strong activation of engaged T cells leads to direct and serial lysis. Furthermore, blinatumomab induces the polyclonal proliferation of activated T cells, which leads to an increased activity of blinatumomab 1 to 2 days after the onset of application [3].

The US Food and Drug Administration (FDA) approved blinatumomab for the treatment of adults and children with B-cell precursor acute lymphoblastic leukemia in first or second complete remission with minimal residual disease (MRD) greater than or equal to 0.1% as well as for relapsed or refractory ALL. The indication according to the European Medicines Agency (EMA) limits the use of blinatumomab to pediatric patients aged one year or older with Philadelphia chromosome negative CD19 positive B-precursor ALL, which is refractory or in relapse after receiving at least two prior therapies or in relapse after receiving prior allogeneic hematopoietic stem cell transplantation (HSCT).

There is still considerable paucity of pediatric data for the use of blinatumomab, and the results of a preponderance of adult trials as well as numerous adult reviews cannot simply be transferred to the pediatric setting. It is widely accepted that pediatric and adult ALL are biologically different with distinct underlying genetic alterations [4,5]. The relapse rate and prognosis are markedly worse in adults [6,7,8] and co-morbidities in adult patients might lead to a different profile of adverse events. Moreover, due to the maturation and expansion of the immune system during the first years of life, lymphocyte subpopulations vary during childhood and differ from adult numbers in relative and absolute size [9], which could affect the activity of blinatumomab.

Rationale for the review: Only four reviews have been published on the use of blinatumomab in the pediatric population [10,11,12,13]. The first three reviews were published several years ago; the very recent review by Shukla and Sulis focuses on high-risk relapsed B-ALL with an excellent summary on the evolvement of treatment in high-risk relapsed ALL. The authors particularly discuss both randomized controlled trials published in the same issue of the *Journal of the American Medical Association* (*JAMA*).

Other pediatric reviews focus on immunotherapy in general [14,15,16], only one of which is less than a year old [17] and a great number of reviews combine adult and pediatric data.

Objective: To provide a comprehensive overview of all published data concerning the use of blinatumomab in children and to summarize all current clinical trials open to pediatric patients.

## 2. Materials and Methods

A PubMed literature search using the terms “blinatumomab and pediatric or children” was performed. The search yielded 127 results by mid-March 2021. In addition, the database ClinicalTrials.gov and EU Clinical Trials Register were searched using blinatumomab and leukemia with “child” as eligibility criteria. The search revealed 25 clinical trials. Four of the clinical trials have been completed and have been published and were, therefore, present in both lists. Of the records screened, 87 were immediately excluded, mostly because they were obviously only focused on adults or on non-Hodgkin lymphoma; we also excluded all manuscripts on nursing and drug preparation as well as all review articles. Of the 61 full-text articles and studies assessed for eligibility, several case reports were excluded because they described single adult patients with adverse events or because the original articles did not contain any clinical information (basic research). Please see the PRISMA flow diagram in Figure 1.

## 3. Results

### 3.1. Efficacy

The first descriptions of the use of blinatumomab in the pediatric population were two small case series of patients with a relapse of ALL after allogeneic HSCT. Handgretinger and colleagues showed that complete remission (CR) after blinatumomab-induced donor T-cell activation in three pediatric patients with post-transplant relapsed ALL was possible [19]. In an extended investigation three years later, nine patients treated with blinatumomab for relapse post-HSCT were analyzed of which six achieved a CR and three did not respond [20].

Until two years ago, there was only one phase I/II study published for the treatment of R/R ALL in pediatric patients (open-label, single-arm phase I/II study at 26 European and US centers NCT01471782). Patients included had refractory or relapsed ALL with >25% bone marrow blasts. The authors showed that blinatumomab clearly demonstrated anti-leukemic activity as a single agent in children with R/R-ALL: among the 70 patients who received the recommended dosage, 27 (39%; 95% CI, 27% to 51%) achieved complete remission within the first two cycles, 14 (52%) of whom achieved complete minimal residual disease response [21]. Furthermore, the follow-up study showed that allogeneic HSCT before or after blinatumomab was associated with a positive effect on survival [22]. In a post hoc analysis day 15 bone marrow minimal residual disease (MRD) predicted complete MRD response to blinatumomab within the first two treatment cycles so that patients with BM MRD ≥ 10^−^^4^ at day 15, being predictive of survival, could potentially pursue alternative therapies, such as dose escalation or combination therapies, to achieve deeper remission [23].

Locatelli and colleagues compared the efficacy of blinatumomab from the single-arm, phase I/II study with that of historical standard of care (SOC) therapy in comparison with three historical comparator groups from North America, Australia and Europe. Single-agent blinatumomab treatment was associated with longer overall survival (OS) and a trend for higher CR in comparison with SOC chemotherapy [24].

Simultaneously, results on the safety and efficacy of blinatumomab in an open-label, single-arm, expanded access international study of pediatric patients with CD19-positive R/R BCPALL were published (RIALTO trial, NCT02187354) [25]. In contrast to the first open label study, patients with a lower tumor burden (≥5% blasts or <5% blasts but with MRD level ≥ 10^−3^) were eligible. Of the 110 patients in the study, 69 patients had CR as the best response in the first two cycles; of these, 45 (65%) proceeded to allogeneic HSCT. There was a trend toward improved OS and relapse-free survival (RFS) for patients who received allogeneic HSCT after blinatumomab compared with those who did not. Median OS for all patients (*n* = 110) was 13.1 months (95% CI 10.2–21.3), with a median follow up of 17.4 months. For all patients reaching or maintaining CR in the first two cycles of blinatumomab (*n* = 69), median RFS was 8.5 months (95% CI 4.4—not evaluable), with a median follow up of 11.2 months.

Over the last two to three years, there have been several case series, single-center experiences or single-country evaluations on the use of blinatumomab in children with R/R ALL. Mouttet and colleagues encouragingly described durable remissions in nine patients with TCF3-HLF-positive ALL, most of whom were treated early in the first consolidation with blinatumomab as a bridge to HSCT [26]. This rare subtype of childhood ALL is usually characterized by a high rate of treatment failure, despite treatment intensification and HSCT. Similarly, Keating and colleagues describe 15 patients in which blinatumomab given prior to transplant reduces MRD and results in favorable leukemia-free survival, toxicity and overall survival [27]. 

Blinatumomab has also successfully been used to treat patients with a high risk for chemotherapy-related adverse events, such as a patient with Down syndrome [28] or patients who experienced overwhelming chemotherapy-associated toxicity during induction therapy. In these patients, blinatumomab served as a bridge to further cytostatic therapy [29]. Infants with ALL are another vulnerable group; the leukemias often harbor *KMT2A* rearrangements and have a high risk for treatment failure and relapse. The last international Interfant trials unfortunately could not improve the outcome of patients below one year of age [30,31]. Clesham and colleagues report on 11 patients with *KMT2A*-rearranged infant ALL [32]. Nine patients became MRD negative, and two patients had a >1-log reduction in MRD prior to HSCT. Three-year EFS and OS post-transplant were 47% and 81%, respectively, comparing favorably with historical outcomes in this subgroup of patients. Four patients relapsed, one of which was MRD-positive pretransplant. One patient relapsed with lineage-switch to monoblastic acute myeloid leukemia and died shortly after.

Colleagues from Spain describe 27 patients treated with blinatumomab and/or inotuzumab, demonstrating that both immunotherapies can induce deep remissions, and blinatumomab can serve as an effective bridging therapy during severe infections [33]. Colleagues from Greece published their experience with nine patients with R/R ALL [34]. They observed a response with morphological CR in 6/9 patients (66.7%) after one cycle of blinatumomab. A successful bridging to HSCT was feasible in 5/9 patients (55.6%), but the median RFS and OS remained low (3.0 and 8.7 months, respectively). Correspondingly, colleagues in Japan conducted an open-label phase 1b study in nine patients [35]. No dose-limiting toxicities were reported; morphological remission within the first two treatment cycles was 56%; one patient had a minimal residual disease response. We described our own single-center experience in 38 patients with R/R-ALL in Tübingen and observed a response to blinatumomab in 13/38 patients (34%) [36]. To date, nine patients (24%) are alive and in complete molecular remission with a median follow-up time of 54 months (8.9–113 months). All survivors underwent haploidentical hematopoietic stem cell transplantation after treatment with blinatumomab. Sutton and colleagues very recently published the Australian experience with blinatumomab in children with R/R-ALL and high-risk genetics [37]. Overall, MRD response was 58%, median follow up was 26 months (14–42 months), 83% proceeded to HSCT and inferior progression-free survival (PFS) was associated with MRD positivity and *KMT2A*-rearranged leukemia.

In March 2021, the first results of two randomized controlled trials investigating blinatumomab in pediatric patients with ALL were published back to back in *JAMA*. Brown and colleagues describe the effect of postreinduction therapy consolidation with blinatumomab versus chemotherapy in patients with first relapse of ALL [38]. All patients received a 4-week reinduction chemotherapy course, followed by randomized assignment to receive two cycles of blinatumomab or two cycles of multiagent chemotherapy, each followed by HSCT. The Children’s Oncology Group conducted this randomized phase 3 clinical trial at hospitals in the US, Canada, Australia and New Zealand (NCT02101853). Eligible patients included those aged 1 to 30 years with B-ALL first relapse. Among 208 randomized patients (median age, 9 years; 97 [47%] females), 118 (57%) completed the randomized therapy. Randomization was terminated at the recommendation of the data and safety monitoring committee without meeting stopping rules for efficacy or futility due to a concern of loss of clinical equipoise. The blinatumomab group presented obvious advantages, such as improved disease-free and overall survival, higher rate of negative MRD and lower rates of serious adverse events. With 2.9 years of median follow up, 2-year disease-free survival was 54.4% for the blinatumomab group vs. 39.0% for the chemotherapy group (hazard ratio for disease progression or mortality, 0.70 [95% CI, 0.47–1.03]; 1-sided *p* = 0.03). Two-year overall survival was 71.3% for the blinatumomab group vs. 58.4% for the chemotherapy group (hazard ratio for mortality, 0.62 [95% CI, 0.39–0.98]; 1-sided *p* = 0.02). In conclusion, postreinduction treatment with blinatumomab did not result in a statistically significant difference in disease-free survival, but the differences between the blinatumomab group and the chemotherapy group in overall survival (71.3% vs. 58.4%) and MRD negativity (75% vs. 32%) were both statistically significant.

The second trial is reported by Locatelli and colleagues [39]. Centers in Europe, Australia and Israel enrolled 108 children older than 28 days and younger than 18 years with high-risk first-relapse B-ALL in morphologic complete remission (M1 marrow, <5% blasts) or with M2 marrow (blasts ≥ 5% and <25%) at randomization: (NCT02393859). Patients were randomized to receive one cycle of blinatumomab or chemotherapy for the third consolidation. A total of 108 patients were randomized, and all patients were included in the analysis. Enrollment was terminated early because it met a prespecified stopping criterion for superiority of the blinatumomab group. After a median of 22.4 months of follow up (IQR, 8.1–34.2), the incidence of events in the blinatumomab vs. consolidation chemotherapy group was 31% vs. 57% (log-rank *p* < 0.001; hazard ratio [HR], 0.33 [95% CI, 0.18–0.61]). Deaths occurred in eight patients (14.8%) in the blinatumomab group and 16 (29.6%) in the consolidation chemotherapy group. The overall survival HR was 0.43 (95% CI, 0.18–1.01). Minimal residual disease remission was observed in more patients in the blinatumomab vs. consolidation chemotherapy group (90% [44/49] vs. 54% [26/48]), and more patients in the blinatumomab group were able to proceed to HSCT. Among children with high-risk first-relapse B-ALL, treatment with one cycle of blinatumomab compared with standard intensive multidrug chemotherapy before allogeneic HSCT resulted in an improved EFS at a median of 22.4 months of follow up. The benefit of blinatumomab was observed in all analyzed subgroups and was especially noticeable for patients with an early relapse.

Please see Table 1 for a list of publications concerning blinatumomab in ALL.

### 3.2. Adverse Events

Cytokine release syndrome (CRS) and neurotoxicity are the most feared adverse events under therapy with blinatumomab. The first pediatric study by von Stackelberg and colleagues described cytopenias as being by far the most common adverse events, obviously mostly preexisting in patients with R/R-ALL. Cytokine release syndrome (CRS) of higher grades was only seen in 4/70 (6%) patients, and 17 patients had neurologic/neuropsychiatric events, mostly tremor, dizziness and somnolence. In nine patients (13%), neurologic events were considered to be treatment related. All events were of grade 2 and resolved; two patients interrupted treatment due to grade 2 seizures, but there were no permanent discontinuations caused by neurologic events [21].

Our own single-center retrospective evaluation showed that cytopenias and febrile reactions were the most common adverse events. Half of the patients experienced CRS, but only 7/38 (18%) of grade ≥3 [36]. High grades of CRS were especially seen in patients who did not receive steroid premedication before blinatumomab, and we demonstrated a clear association between high tumor load and the development of CRS. Neurotoxicity was seen in seven patients (18%); only two patients discontinued blinatumomab therapy due to generalized seizures. 

A case series in 11 infants described three patients with grade 1-2 CRS and one patient with neurotoxicity (confusion and somnolence); symptoms were resolved by interrupting the blinatumomab infusion [32].

In newly published RCTs, Brown and colleagues describe blinatumomab-related adverse events with overall 22% CRS, 11% encephalopathy and 4% seizures but only one case of grade ≥3 CRS or seizure each and two cases of higher-grade encephalopathy. Other rates of notable serious adverse events were much less common in the blinatumomab group compared to the chemotherapy group: infection (15% vs. 65%), febrile neutropenia (5% vs. 58%), sepsis (2% vs. 27%) and mucositis (1% vs. 28%) [38].

Locatelli and colleagues reported an incidence of serious adverse events of 24.1% vs. 43.1% in a blinatumomab vs. consolidation chemotherapy group. The incidence of adverse events grade ≥3 was also lower in the blinatumomab group (57.4% vs. 82.4%). The most frequently reported adverse events were neurologic symptoms and seizure (each 3.7%) in the blinatumomab group and febrile neutropenia (17.6%) in the consolidation chemotherapy group. Only two patients in the blinatumomab group and one in the consolidation chemotherapy group experienced CRS at less than grade 3.

### 3.3. CD19 Expression

CD19-negative relapses of pediatric BCP-ALL following blinatumomab treatment were first described in 2017 from a phase I/II study: four patients experienced CD19-negative relapse after prior blinatumomab-induced hematologic remission, and one patient showed CD19-negative progression during treatment after 10 days in cycle 1 with blasts showing a monocytic phenotype [40].

In the retrospective evaluation of our own 38 patients in Tübingen, none of the patients displayed a CD19-negative subclone detectable by flow cytometry before receiving a first cycle of blinatumomab. Sixteen patients had CD19-positive relapse. One patient experienced a CD19-negative relapse after the second cycle was completed. Another patient’s leukemia did not express CD19 by flow cytometric analysis during the second cycle. Interestingly, the leukemic cells quickly regained normal CD19 expression after cessation of blinatumomab [36]. One patient with *KMT2A* translocation showed myeloid differentiation in addition to the disappearance of CD19 under treatment with blinatumomab and spontaneous conversion back to a CD19-positive immunophenotype after the discontinuation of blinatumomab. 

A study analyzing immunophenotypic changes in leukemic cells at relapse in 90 pediatric R/R ALL patients treated with blinatumomab showed that in 21 cases, leukemia cells at relapse were CD19 positive, whereas in six cases, they were CD19 negative [43]. Three children (two with *KMT2A* gene rearrangement and one with germline *KMT2A*) developed relapse through lineage switch to CD19-negative acute myeloid leukemia, mixed phenotypic acute leukemia and unclassifiable leukemia. This switch in immunophenotype has previously been described in case reports by others [42,45]. One case report also describes such a lineage switch following blinatumomab in a young girl post-HSCT whose leukemia did not harbor *KMT2A* rearrangement [41].

The Australian group described two patients with CD19-negative relapses, one of which also harbored a *KMT2A* rearrangement and showed myeloid differentiation; however, the leukemias of most patients who relapsed remained CD19 positive [37].

### 3.4. Clinical Trials

Of 78 trials listed on Clinical trills.gov, only 24 include pediatric patients. One trial was only listed in the European registry. All are interventional open-label trials.

In addition to the phase I/II studies in R/R ALL and both recently published phase III randomized controlled trials (RCTs) mentioned above, there are further trials ongoing for patients with a refractory or relapsed leukemia. One study in Japan is still recruiting, with adult data already published [46], but pediatric results still pending (NCT02412306). An observational retrospective study sponsored by Amgen for children and adults with Ph-chromosome-negative R/R ALL was completed, but there are no published results yet (NCT02783651). Checkpoint inhibitors might increase T-cell proliferation and enhance the mechanism of action of blinatumomab. Adolescents and adults with poor-risk R/R ALL are eligible for a trial of the National Cancer Institute where blinatumomab and nivolumab are administered with or without ipilimumab (NCT02879695). Similarly, the Children’s Hospital Medical Center in Cincinnati planned a pilot study to assess the safety, tolerability and preliminary anti-tumor activity of combining pembrolizumab and blinatumomab in children and young adults with R/R ALL (NCT03605589). This study is currently suspended due to slow recruitment, an amendment is pending. Moreover, the National Cancer Institute is conducting a phase II trial in children and young adults with first relapse of ALL comparing blinatumomab alone to blinatumomab with nivolumab.

Many studies are investigating blinatumomab in relation to HSCT, either as a bridging element or as a consolidation treatment afterwards. The Medical College of Wisconsin has two ongoing trials: Blina Part 1 explores blinatumomab as a bridging therapy for patients in first or greater relapse (NCT04556084), and Part 2 is focused on blinatumomab after T-cell receptor (TCR) alpha/beta-depleted HSCT (NCT04746209). The University of British Columbia is planning a trial on blinatumomab for MRD in pre-B-ALL patients following HSCT but is not yet recruiting (NCT04044560). Similarly, Seoul national University is planning a single-arm study for patients with persistent or recurrent MRD before HSCT (NCT04604691) but is currently not recruiting. M.D. Anderson Cancer Center are evaluating blinatumomab maintenance following allogeneic HSCT in children and adults (NCT02807883). Equivalently, European colleagues included an add-on study for blinatumomab post-HSCT into the ALL SCTped 2012 study (NCT04785547). St. Jude Children’s Research Hospital is currently conducting two trials for several hematological malignancies receiving naïve T-cell-depleted haploidentical HSCT. The first study combines TCRgamma/delta T cells and memory T cells with the selected use of blinatumomab in relapsed/refractory malignancies (NCT02790515). The second couples a TCRalpha/beta-depleted progenitor cell graft with an additional memory T-cell donor lymphocyte infusion (DLI), plus the selected use of blinatumomab a week after DLI (NCT03849651).

Blinatumomab has also found its way into the frontline treatment of ALL in children. In the United States, the M.D. Anderson Cancer Center is recruiting adolescents aged 14 years and older for a phase II trial with blinatumomab and combination chemotherapy as frontline therapy (NCT02877303) and patients in the same age group for a phase II trial combining blinatumomab with inotuzumab ozogamicin (NCT02877303). The TOTAL Therapy XVII of St. Jude Children’s Research Hospital is open for children aged 1 to 18 years (NCT03117751). The National Cancer Institute phase III trial includes patients aged 1 to 31 years (NCT03914625). Both studies include patients with Philadelphia-chromosome-positive (Ph+) leukemias. The federal research institute of pediatric hematology, oncology and immunology in Russia are also conducting an interventional trial with one course of blinatumomab in consolidation therapy as the experimental arm (NCT04723342). Simultaneously, several large randomized multicenter phase III trials in Europe are investigating blinatumomab as a frontline treatment. ALLTogether1 is a treatment study protocol for children and young adults (1–45 years of age) using a sequential assignment to therapy (NCT04307576), and AIEOP-BFM ALL 2017 is open for children aged below 18 years using a factorial assignment (NCT03643276). Colleagues from the Netherlands are conducting a pilot study to test the feasibility, safety and efficacy of adding blinatumomab to the Interfant-06 backbone in infants with *KMT2A*(*MLL)*-rearranged ALL (EudraCT 2016-004674-17, no NCT identifier). Together with the AIEOP-BFM ALL 2017 study, these are currently the only frontline randomized clinical trials including infants with ALL.

Please see Table 2 for a list of all clinical trials of blinatumomab in pediatric patients.

## 4. Discussion

The results of the phase I/II study and several single-institution or national retrospective evaluations show that children with R/R-ALL show a response to blinatumomab ranging from 34–38% [21,36] to around 60% [25,37]. In almost all studies published to date, there is evidence or at least a trend of improved survival if blinatumomab is administered prior to or after allogeneic HSCT [22,25,26,32,36].

The results of both phase III RCTs in children with first relapse of ALL confirm the superiority of blinatumomab in achieving MRD-negativity before HSCT and even show evidence for an advantage in overall survival [38,39], while inducing less severe adverse events compared to conventional chemotherapy. These results definitely warrant the inclusion of blinatumomab into relapse protocols before HSCT. 

The prognostic importance of achieving MRD negativity prior to HSCT has been well established [47]. Response rates for blinatumomab in R/R-ALL are encouraging, but they are still insufficient. Survival outcomes for the non-responding patients remain extremely poor [48,49]. Combination therapies with other antibodies, such as inotuzumab ozogamicin, will hopefully overcome problems of CD19-escape (current trial NCT02877303). Administering donor lymphocyte infusions in a haploidentical setting (NCT02790515 and NCT03849651) or adding a checkpoint inhibitor, such as PD-1- or CTLA-4-inhibitors, could enhance the efficacy of blinatumomab (NCT02879695, NCT04546399 and NCT03605589).

Overall, adverse events are much less common under blinatumomab compared to conventional chemotherapy [38,39]. Additionally, even specific blinatumomab-related toxicities, such as CRS and neurotoxicity, only rarely necessitate the interruption of therapy [20,31,35]. This observation makes blinatumomab an ideal drug for selected use in vulnerable patients, such as children with Down syndrome, with a high risk for chemotherapy-related mortality [50] or other patients experiencing overwhelming toxicities [29]. To date, only two case reports have described the benefit of using blinatumomab in Down syndrome children [28,29]. No Down syndrome patients were included in the pediatric RCT for R/R-ALL, but several frontline trials are enabling Down syndrome patients with high-risk features access to upfront treatment with blinatumomab (NCT03643276, NCT04307576 and NCT03117751). This approach might also prove to be a worthwhile strategy in the group of patients with underlying genetic defects, such as chromosomal instability or defects in DNA-damage repair.

On the other hand, patients with ALL harboring almost incurable translocations, such as TCF3/HLF-fusion, are also directly eligible for receiving blinatumomab upfront, as these patients are known to display a high rate of treatment failure with conventional chemotherapy [51]. The promising results of durable remissions in nine patients with TCF3/HLF-positive leukemia confer great hopes [26].

Philadelphia-chromosome-positive (Ph+) patients comprise another important subgroup. Patients with Ph+ leukemia have a poor outcome and are, therefore, treated with high-risk regimens, including a tyrosine kinase inhibitor targeting the BCR/ABL-fusion [52]. Continuous concomitant medication with imatinib or dasatinib leads to an important increase in adverse events, necessitating alternative therapeutic options. Sutton and colleagues observed relatively good outcomes for patients with Philadelphia-positive or Philadelphia-like ALL treated with blinatumomab [37]. Unfortunately, most upfront trials with blinatumomab for children open today do not include patients with Ph+ ALL, with the exception of Total Therapy XVII (NCT03117751).

Infants with ALL are a unique subgroup of patients; infant leukemias often harbor *KMT2A* rearrangement and also display high relapse rates and unsatisfying outcomes [30,31]. Clesham and colleagues described 11 infants treated with blinatumomab; treatment was well tolerated, and complete MRD responses were seen in the majority of cases. All children received HSCT, and the 12-month EFS compares favorably with historical outcomes [32]. In contrast, in an Australian study, infants with *KMT2A*-rearranged leukemia had poor outcomes with an MRD response rate of only 44% [37]. This disparity might be explained by pretreatment or tumor burden differences prior to treatment. One European study is currently adding blinatumomab treatment to the Inferfant-06 backbone (EudraCT 2016-004674-17), and the frontline AIEOP-ALL BFM trial also includes infants (NCT03643276). The results will hopefully clarify the utility of blinatumomab in these patients, and also regarding the lineage switch with the outgrowth of myeloid leukemia that several groups have described for *KMT2A*-rearranged leukemias under the selective pressure of blinatumomab [36,42,45]. The combination of targeting two antigens might be a strategy in such cases. A recent case report describes the successful combination of blinatumomab with gemtuzumab ozogamicin in an infant with *KMT2A*-rearraged mixed phenotypic leukemia [44].

The first mention of blinatumomab in three pediatric patients was published ten years ago in 2011 [19], followed by the description of nine children receiving blinatumomab after relapse post-HSCT [20]. It took a further two years until the publication of the phase I/II trial [21]. The first results of RCTs for the use of blinatumomab in children with first relapse of ALL were published this year [38,39]. Current front-line trials are investigating blinatumomab in children with standard-risk ALL (NCT03914625, NCT02877303); others only apply blinatumomab to standard-risk patients with residual disease at the end of induction therapy (NCT03117751) or intermediate and high-risk ALL (NCT0363276). Results from these trials will show whether replacing part of the classic chemotherapy with blinatumomab is feasible without impairing EFS. The prognosis of standard-risk leukemia patients is excellent; risk-stratified therapy has reduced late morbidity and mortality [53]. Direct treatment-related morbidity and mortality, especially due to infections, cardio-metabolic dysfunction, hepatotoxicity, osteonecrosis and asparaginase-associated problems, such as coagulation disorders and pancreatitis, remain an important issue. Replacing steroids and cytostatic drugs with immunotherapeutics, such as blinatumomab, might help reduce these problems.

Compared to treatment with chimeric antigen receptor (CAR) T cells, blinatumomab has a few advantages: it is an off-the-shelf product; it is less expensive; the short half-life enables the precise control of serum levels; and a quick reduction is possible in the case of an adverse event, such as neurotoxicity or CRS. Outpatient delivery by a portable minipump enables a good health-related quality of life. The direct comparison of blinatumomab with CAR T cells or other immunotherapies, such as inotuzumab ozogamicin, are lacking, and further studies are necessary to help determine at which point each therapeutic option might yield the best results. However, it has been shown that sequential treatment is often feasible, in both directions [54,55].

## 5. Conclusions

Patients in currently published studies and case series have all been heavily pretreated, and today, blinatumomab has evolved to being a first-line salvage therapy in many centers, but there are still only two RCTs published with results in the pediatric setting for the use of blinatumomab in first relapse or even in a situation with rising MRD levels. All available data in R/R-ALL suggest a necessity for HSCT after a bridging therapy with blinatumomab. Ongoing trials will show whether blinatumomab is capable of inducing lasting remissions without a following allogeneic HSCT or constitutes a suitable maintenance therapy post-HSCT. Adult data suggest that not all MRD responders necessarily require a transplant [56].

Many questions remain unanswered. Some have recently been pointed out by Shukla and Sulis [13], such as the following: when is the optimal time to introduce blinatumomab, and how many cycles are needed? Current frontline protocols have different approaches (blinatumomab after or instead of consolidation therapy; one versus two cycles). Does blinatumomab have a value as a consolidation treatment in non-high-risk patients with negative MRD in terms of reducing the relapse rate? Why do some patients respond and others do not? This is definitely not merely due to the loss of the target antigen CD19.

Ongoing trials will hopefully clarify these questions in the near future.

## Figures and Tables

**Figure 1 jcm-10-02544-f001:**
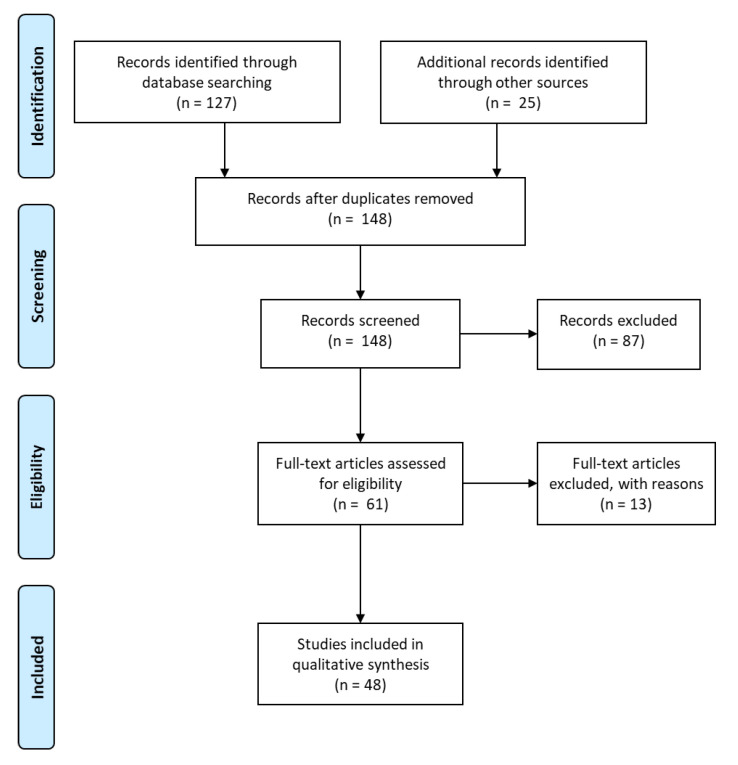
PRISMA flow chart [18].

**Table 1 jcm-10-02544-t001:** Table with all articles published on use of blinatumomab in pediatric ALL.

Author	Year	Ref. ^1^	Patients	Title
Handgretinger	2011	[19]	3 R/R-ALL patients post-HSCT	CR after blinatumomab-induced donor T-cell activation in three pediatric patients with post-transplant relapsed ALL
Schlegel	2014	[20]	9 R/R-ALL patients post-HSCT	Pediatric post-transplant R/R BCP ALL leukemia shows durable remission by therapy with the T-cell engaging bispecific antibody blinatumomab
Von Stackelberg	2016	[21]	93 R/R patients (70 with recommended dosage)	Phase I/phase II study of blinatumomab in pediatric patients with R/R ALL
Mejstríková	2017	[40]	18 patients (4 with CD19-negative relapse)	CD19-negative relapse of pediatric BCP-ALL following blinatumomab treatment
Zoghbi	2017	[41]	case report	Lineage switch under blinatumomab treatment of relapsed common ALL without MLL rearrangement
Wadhwa	2018	[28]	case report	Blinatumomab activity in a patient with Down syndrome BCP-ALL
Gore	2018	[22]	70 R/R-ALL patients	Survival after blinatumomab treatment in pediatric patients with R/R BCP-ALL
Wölfl	2018	[42]	case report	Spontaneous reversion of a lineage switch following an initial blinatumomab-induced ALL-to-AML switch in MLL-rearranged infant ALL
Mouttet	2019	[26]	9 TCF3/HLF	Durable remissions in TCF3-HLF positive acute lymphoblastic leukemia with blinatumomab and SCT
Keating	2019	[27]	15 children MRD-positive before HSCT	Reducing minimal residual disease with blinatumomab prior to HSCT for pediatric patients with ALL
Elitzur	2019	[29]	11 pediatric patients with overwhelming chemotherapy-associated toxicity	Blinatumomab as a bridge to further therapy in case of overwhelming toxicity in pediatric BCP-ALL
Brown	2019	[23]	59 patients of the MT103-205 study (NCT01471782)	Day 15 bone marrow MRD predicts response to blinatumomab
Locatelli	2020	[24]	70 patients of the MT103-205 study (NCT01471782)	Blinatumomab versus historical standard therapy in pediatric patients with R/R Ph-negative BCP-ALL
Locatelli	2020	[25]	110 R/R-ALL patients	Blinatumomab in pediatric patients with R/R ALL: results of the RIALTO trial, an expanded access study
Mikhailova	2020	[43]	90 patients	Immunophenotypic changes of leukemic blasts in children with R/R- ALL who have been treated with blinatumomb
Contreras	2020	[33]	27 children/young adults treated with blinatumomab and/or inotuzumab	Clinical utilization of blinatumomab and inotuzumab immunotherapy in children with relapsed or refractory B-ALL
Clesham	2020	[32]	11 infants	Blinatumomab for infant ALL
Horibe	2020	[35]	9 children	A phase 1 study of blinatumomab in Japanese children
Ampatzidou	2020	[34]	9 children	Insights from the Greek experience of the use of Blinatumomab in pediatric R/R ALL
Queudeville	2021	[36]	38 R/R-ALL patients	Blinatumomab in pediatric patients with relapsed/refractory B-cell precursor acute lymphoblastic leukemia
Sutton	2021	[37]	24 R/R-ALL patients outside of clinical trials	Outcomes for Australian children with relapsed/refractory acute lymphoblastic leukaemia treated with blinatumomab
Brethon	2021	[44]	case report	Targeting 2 antigens as a promising strategy in mixed phenotype acute leukemia: combination with blinatumomab with gemtuzumab ozogamicin in an infant with KMT2A-rearraged leukemia
Brown	2021	[38]	208 pts, 1 to 30 years	Effect of Postreinduction Therapy Consolidation with Blinatumomab vs. Chemotherapy on Disease-Free Survival in Children, Adolescents, and Young Adults with First relapse of B-Cell Acute Lymphoblastic Leukemia NCT02101853
Locatelli	2021	[39]	108 pts, 28 days to 18 years	Effect of Blinatumomab vs. Chemotherapy on Event-Free Survival Among Children with High-risk First-Relapse B-Cell Acute Lymphoblastic Leukemia: A Randomized Clinical Trial NCT02393859

^1^ References according to mention in this article.

**Table 2 jcm-10-02544-t002:** Table containing all ongoing clinical trials with blinatumomab for pediatric patients.

Clinical Trials Identifier	Other Study ID Numbers	Ref. ^1^	Title	Age	Status
NCT01471782	MT103-205 2010-024264-18 (Eudra-CT)	[21]	Clinical Study With Blinatumomab in Pediatric and Adolescent Patients With Relapsed/Refractory B-precursor Acute Lymphoblastic Leukemia	Up to 17 years (child)	completed
NCT02187354	RIALTO 2014-001700-21 (EudraCT)	[25]	Expanded Access Protocol-Blinatumomab in Pediatric & Adolescent Subjects with Relapsed/Refractory B-precursor ALL (RIALTO)	Up to 17 years (child)	completed
NCT02783651	20150253		A Study of Patients with Ph- Chromosome-negative Relapsed or Refractory Acute Lymphoblastic Leukemia in the US	Child, adult	completed
NCT02879695	NCI-2016-01300 (CTRP)		Blinatumomab and Nivolumab With or Without Ipilimumab in Treating Patients With Poor-Risk Relapsed or Refractory CD19+ Precursor B-Lymphoblastic Leukemia	16 years and older	recruiting
NCT02393859	2014-002476-92 (EudraCT)	[39]	Phase 3 Trial of Blinatumomab vs. Standard Chemotherapy in Pediatric Subjects With High-Risk (HR) First Relapse B-precursor Acute Lymphoblastic Leukemia (ALL)	Up to 17 years (child)	active, not recruiting
NCT04546399	NCI-2020-06813 (CTRP)		A Study to Compare Blinatumomab Alone to Blinatumomab With Nivolumab in Patients Diagnosed With First Relapse B-Cell Acute Lymphoblastic Leukemia (B-ALL)	1 to 31 years (child, adult) Including Down syndrome patients	recruiting
NCT03914625	NCI-2019-02187 (CTRP)		A Study to Investigate Blinatumomab in Combination With Chemotherapy in Patients With Newly Diagnosed B-Lymphoblastic Leukemia	1 to 31 years (child, adult) Including Down syndrome patients	recruiting
NCT02101853	NCI-2014-00631 (CTRP) COG-AALL1331	[38]	Blinatumomab in Treating Younger Patients With Relapsed B-cell Acute Lymphoblastic Leukemia	1 to 31 years (child, adult)	Active, not recruiting
NCT02877303	NCI-2017-00596 (CTRP)		Blinatumomab and Combination Chemotherapy as Frontline Therapy in Treating Patients With B Acute Lymphoblastic Leukemia	14 years and older	recruiting
NCT02790515	REF2HCT NCI-2016-00812 (CTRP)		Provision of TCRγδ T Cells and Memory T Cells Plus Selected Use of Blinatumomab in Naïve T-cell Depleted Haploidentical Donor Hematopoietic Cell Transplantation for Hematologic Malignancies Relapsed or Refractory Despite Prior Transplantation	Up to 21 years	recruiting
NCT03849651	HAP2HCT		TCRαβ-depleted Progenitor Cell Graft With Additional Memory T-cell DLI, Plus Selected Use of Blinatumomab, in Naive T-cell Depleted Haploidentical Donor Hematopoietc Cell Transplantation for Hematologic Malignancies	Up to 21 years	recruiting
NCT04307576	ALLTogether1 2018-001795-38 (EudraCT)		ALLTogether1—A Treatment Study Protocol of the ALLTogether Consortium for Children and Young Adults (1–45 Years of Age) With Newly Diagnosed Acute Lymphoblastic Leukaemia (ALL)	1 to 45 years	recruiting
NCT03643276	AIEOP-BFM ALL 2017 2016-001935-12 (EudraCT)		Treatment Protocol for Children and Adolescents With Acute Lymphoblastic Leukemia-AIEOP-BFM ALL 2017	Up to 17 years	recruiting
NCT03117751	TOT17 NCI-2017-00582 (CTRP)		Total Therapy XVII for Newly Diagnosed Patients With Acute Lymphoblastic Leukemia and Lymphoma	1 to 18 years	recruiting
	2016-004674-17 (EudraCT) ML59901.078.17		A pilot study to test the feasibility, safety and efficacy of the addition of the BiTE antibody Blinatumomab to the Interfant-06 backbone in infants with MLL-rearranged acute lymphoblastic leukemia. A collaborative study of the Interfant network	Up to 17 years	recruiting
NCT04604691			Blinatumomab in Pediatric B-cell Acute Lymphoblastic Leukemia (ALL) with Minimal Residual Disease (MRD)		Not yet recruiting
NCT03605589			Pembro and Blina combination in Pediatric and Young Adult Patients With Relapsed/Refractory Acute Leukemia or Lymphoma	1 to 40 years	Suspended, slow recruitment, amendment pending
NCT04723342	ALL-MB 2019 Pilot		Treatment of Children and Adolescents With Primary B-precursor Acute Lymphoblastic Leukemia With Combination Chemotherapy and Immunotherapy	1–18 years	recruiting
NCT04556084	Blina Part 1		Blinatumomab Bridging Therapy	Up to 25 years	recruiting
NCT04746209	Blina Part 2		Blinatumomab After TCR Alpha Beta/CD19 Depleted HCT	Up to 25 years	not yet recruiting
NCT02807883	NCI-2016-01182		Blinatumomab Maintenance Following Allogeneic Hematopoietic Cell Transplantation for Patients With Acute Lymphoblastic Leukemia	1 to 70 years	active, not recruiting
NCT02412306	20130265		Study of Blinatumomab in Japanese Patients With Relapsed/Refractory B-precursor Acute Lymphoblastic Leukemia	<18 years for pediatric subjects	recruiting
NCT02877303	2014-0845 NCI-2017-00596		Blinatumomab, Inotuzumab Ozogamicin, and Combination Chemotherapy as Frontline Therapy in Treating Patients With B Acute Lymphoblastic Leukemia	14 years and older	recruiting
NCT04044560	H19-00893 CTTC 1902		Blinatumomab for MRD in Pre-B-ALL Patients Following Stem Cell Transplant (OZM-097)	1 year and older (children and adults)	not yet recruiting
NCT04785547	FORUM Add-on Blina post TX		ALL SCTped 2012 FORUM Add-on Study Blina Post HSCT	6 months to 21 years	recruiting

^1^ References according to mention in this article.

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
