# Peer review of "Blinatumomab in Pediatric Acute Lymphoblastic Leukemia—From Salvage to First Line Therapy (A Systematic Review)"

_jcm, 2021, doi:10.3390/jcm10122544_

Round 1

Reviewer 1 Report

The review Blinatumomab in pediatric acute lymphoblastic leukemia  from salvage to first line therapy (a systematic review) by Queudeville et al. presents a good analysis of the employment of Blinatumomab in the current clinical practice in pediatric patients.

There is a good description of the current employment in patients for MRD, fragile its, pre and post HSCT and current clinical trials

Discussion and Conclusions critically analyze the current, the correct and the future directions of the treatment with blinatumomab.

The paragraphs are sometimes too Lon and the reading can result confusing.

Some sentences need to be fixed in order to make the paper easier to understand.

The language should be fixed.

Below find some of the major sentences that need to be revised.

Please check this period:

Line 137 "the last international trials unfortunately could not improve outcome of these small patients [2  " Iwould notes "small" and the sentence should be fixed

Line 148 I would use observed instead of saw

Line 164

"Brown and colleagues describe the effect of post-reinduction therapy consolidation with 165 blinatumomab versus chemotherapy in patients with first relapse of ALL [37]" Please clarify this period 

Line 168 -176 Please check this paragraph.Sentences are too long and the period results difficult to read.

line 211 Please check language under administration..."

Line 219 Our own single-center retrospective evaluation also showed a large number of cytopenias as well as high frequency of febrile reactions in general. Please check language. The sentence should be simplified and fixed

line 226 -227

The case-series in 11 infants described three patients with grade 1-2 CRS and one 226 patient with neurotoxicity (confusion and somnolence), which resolved on interrupting the infusion [31].. Please check language

line 243

"When targeting a surface antigen, the monitoring of CD19-expression in case of non- response or relapse is of great importance"

Please check language or remove

line 252 one patient’s leukemia lacked CD19-expression between the second and third cycle. Do you mean "CD19 expression was not detected by flow cytometry in one patient, and was subsequently quickly re-observed

Line 258 Please check language

An immunophenotypic study conducted in 90 pediatric R/R ALL patients treated with blinatumomab showed that in 21 cases leukemia cells at relapse were CD19-positive 259 whereas in six cases they were CD19-negative 

line 263 This phenomenon has previously described in case reports by others . Please fix this sentence

line 277

For the R/R-ALL patients one idea of further enhancing the mechanism 278 of action of blinatumomab is by combining it with checkpoint inhibitors. Please check language and clarify

line 334 proceeded or 334 followed by allogeneic HSCT. Please check language and meaning

Line 342-343 And while the response rates for blinatumomab in R/R-ALL are encouraging they are still insufficient as survival outcomes for the non-responding patients remain extremely poor. Please clarify this sentence

Author Response

We thank the reviewer for the very thorough scrutiny of our work. We addressed all remarks as noted below.

Line 137: The long sentence was split into two separate sentences, the name of the Interfant trials was inserted. “Small” was replaced by “patients below one year of age”.

Infants with ALL are another vulnerable group, the leukemias often harbor KMT2A-rearrangements and have a high risk for treatment failure and relapse. The last international Interfant trials unfortunately could not improve outcome of patients below one year of age  [1, 2].

Line 148: “Saw” was replaced by “observed”.

Line 164: “Brown and colleagues…”. The reviewer asks to please clarify this period. The details of the study were explained two sentences later. To make the paragraph more easily understandable, we changed the order of the sentences.

Brown and colleagues describe the effect of post-reinduction therapy consolidation with blinatumomab versus chemotherapy in patients with first relapse of ALL [3].  All patients received a 4-week reinduction chemotherapy course, followed by randomized assignment to receive 2 cycles of blinatumomab or 2 cycles of multiagent chemotherapy, each followed by HSCT. The Children's Oncology Group conducted this randomized phase 3 clinical trial at hospitals in the US, Canada, Australia, and New Zealand (NCT02101853).

Lines 168-176: The paragraph describing the paper by Brown and colleagues was rewritten, sentences were shortened.

Randomization was terminated at the recommendation of the data and safety monitoring committee without meeting stopping rules for efficacy or futility because of a concern of loss of clinical equipoise. The blinatumomab group presented obvious advantages, such as better disease-free and overall survival, higher rate of negative MRD and lower rates of serious adverse events.

Line 211: “under administration of” was replaced by “under therapy with blinatumomab”

Line 219: The sentence was rewritten and simplified.

Our own single-center retrospective evaluation showed that cytopenias and febrile reactions were the most common adverse events.

Line 226-227: The sentence was rewritten.

A  case-series in 11 infants described three patients with grade 1-2 CRS and one patient with neurotoxicity (confusion and somnolence), symptoms  resolved by  interrupting the blinatumomab infusion [4].

Line 243: The sentence was removed.

Line 252: The misleading formulation was improved.

One patient experienced a CD19-negative relapse after the second cycle was completed. Another patient’s leukemia did not express CD19 by flow cytometric analysis during the second cycle. Interestingly, the leukemic cells quickly regained normal CD19-expression after cessation of blinatumomab  [5].

Line 258: The sentence was rephrased.

A study analyzing immunophenotypic changes of leukemic cells at relapse in 90 pediatric R/R ALL patients treated with blinatumomab showed that in 21 cases leukemia cells at relapse were CD19-positive whereas in six cases they were CD19-negative [6].

Line 263: The sentence was corrected.

This switch in immunophenotype has previously been described in case reports by others [7, 8].

Line 277: The sentence was rewritten and clarified:

Checkpoint inhibitors might increase T-cell proliferation and enhance the mechanism of action of blinatumomab.

Line 334: The sentence was corrected:

In almost all studies published to date there is evidence or at least a trend for better survival if blinatumomab is given prior to or after  allogeneic HSCT [4, 5, 9-11].

Lines 342-343: The sentence was clarified:

Response rates for blinatumomab in R/R-ALL are encouraging but they are still insufficient. Survival outcomes for the non-responding patients remain extremely poor [12, 13].

The lines indicated are the lines initially indicated by the reviewer in his/her comments. Due to the changes in the manuscript the current line numbers have changed.

Reviewer 2 Report

Comprehensive review, very well written. Gives  complete up to date information about published results and opened protocols. The language is of good quality.

The only question which should be additionally discussed is why 10 years after first publication of the use of blinatumomab in pediatric patients we still do not have precise data concerning the use of the drug in first line protocols. Taking into account the efficacy of blina, it probably could replace the classic chemo in low and intermediate risk groups. Maybe it is time to organize transatlantic randomized trials to explain the role of blinatumomab in modern ALL protocols.

Author Response

Reviewer 2 asked us to additionally discuss why data on use of blinatumomab in first line protocols is still lacking. We absolutely agree on the necessity to evaluate blinatumomab as a frontline therapy in standard and intermediate risk patients. We included the following paragraph into the discussion:

The first mention of blinatumomab in three pediatric patients was published ten years ago in 2011 [14], followed by the description of nine children receiving blinatumomab after relapse post-HSCT [15]. It took further two years until publication of the phase I/II trial [16]. First results of RCT for use of blinatumomab in children with first relapse of ALL were published this year [3, 17]. Current front-line trials are investigating blinatumomab in children with standard risk ALL (NCT03914625, NCT02877303), others only apply blinatumomab to standard risk patients with residual disease end of induction (NCT03117751) or intermediate and high risk ALL (NCT0363276). Results from these trials will show whether replacing part of the classic chemotherapy by blinatumomab is feasible without impairing EFS and OS. The prognosis of standard risk leukemia patients is excellent, risk-stratified therapy has reduced late morbidity and mortality [18]. Direct treatment-related morbidity and mortality, especially due to infections, cardio-metabolic dysfunction, hepatotoxicity, osteonecrosis and asparaginase-associated problems such as coagulation disorders and pancreatitis remain an important issue. Replacing steroids and cytostatic drugs with immunotherapeutics such as blinatumomab might help reduce these problems.

Reviewer 3 Report

I would like to thank the authors for this manuscript on the use of Blinatumomab in pediatric ALL with a systematic  review. Thank you for giving me the opportunity to read this paper and express my comments

The use of Blinatumomab in childhood R/R ALL has clearly showed promising results expecially as a bridge to HSCT. Less toxicity has been reported particularly in heavily pre-treated patients allowing these patients to better proceed to HSCT (less toxicity and lower MRD levels compared with intensive chemotherapy). Based on R/R ALL results, also reported by randomized trials, today Blinatumomab is used in first-line protocols for pediatric patients at high-risk. The results of the ongoing trials will probably confirm the activity of this drug and also, show whether blinatumomab must be followed by transplant or can by itself ensure prolonged remissions, as reported in some adult patients.

However, so far, the optimal time to introduce Blinatumomab in both first-line therapies and salvage protocols, and how many cycles are needed, still needs to be clarified.

Also, the role of Blinatumomab for extramedullary relapse (especially CNS) or for those patients with extramedullary involvement at disease diagnosis is still unclear. Could the authors make any suggestions? Are there data on these patients who have received Blinatumomab? In particular, both in the first line of high-risk patients, and in relapse, is high-dose systemic chemotherapy required (before Blinatumomab) to reduce the risk of further recurrence in extramedullary sites?

Furthermore, the use of Blinatumomab for pediatric Ph + ALL is of great interest, even if related to a few numbers of cases. Combination of blinatumomab with tyrosine kinase inhibitors has shown promising results in adults with Ph+ALL; in pediatric age, high-dose chemotherapy and tyrosin kinase inhibitors are improving the outcome of patients but toxicity is relevant. Sutton has recently reported good results in few Ph+ and Ph-like pediatric ALL; some ongoing trials have included these childrenin the randomized Blinatumomab arms. I ask if the authors have any further information regarding children with Ph+/Ph-like ALL who have already received blinatumomab; if yes, is it possible to comment on the outcome?

Author Response

We thank reviewer 3 for raising two pivotal questions which both remain largely unanswered by current publications. We did not comment on both issues in our review because of our focus on pediatric data. We here report all relevant findings in adults and would be happy to include them into the manuscript if desired by reviewers and editors.

Extramedullary manifestations

Extramedullary manifestations (EM) are common in ALL, especially infiltration of the central nervous system (CNS).

In the prescribing information of blinatumomab by Amgen they state: There is limited experience with BLINCYTO in patients with active ALL in the central nervous system (CNS) or a history of neurologic events. Patients with a history or presence of clinically relevant CNS pathology were excluded from clinical studies.

Even in CNS leukemia cleared by i.th. chemotherapy or irradiation there remains the concern that these patients could experience enhanced neurotoxic adverse events. In our own retrospective study CNS-disease had always been cleared through intrathecal administration of chemotherapy or radiotherapy prior to administration of blinatumomab. Of all six patients with previous CNS manifestation of the leukemia, only one developed neurotoxicity under blinatumomab and it merely occurred during the second cycle [5].

In the recent RCT by Locatelli et al. a total of 24 patients with history of extramedullary involvement (19 with prior CNS involvement, 2 testis and 3 others) were included. A diagnostic lumbar puncture was performed before randomization and patients with evidence of CNS involvement were excluded. The authors themselves state that the numbers of patients who experienced an isolated extramedullary involvement at first high-risk relapse was too low to allow inference of definitive conclusions on the efficacy of blinatumomab in comparison with consolidation chemotherapy in this patient subset.

Similarly, the other recent RCT by Brown et al. included a total of 20 patients with isolated extramedullary disease. Patients with residual CNS leukemia after reinduction were not eligible for randomization.

There is only one case report of a young man with second relapse of CNS leukemia receiving three weeks of intrathecal chemotherapy directly succeeded by blinatumomab and further intrathecal chemotherapy in parallel [19]. The patient developed central nervous system toxicity with speech impairment, altered mental state, incontinence and weakness. After discontinuation of blinatumomab and administration of dexamethasone he gradually recovered.

There is reason to fear additive neurotoxicity due to concomitant use of blinatumomab and intrathecal chemotherapy, especially in patients with CNS leukemia or other underlying CNS disease. The routine administration of one prophylactic dose of intrathecal chemotherapy for CNS negative patients directly before start of blinatumomab is used in most protocols and does not seem to enhance neurotoxicity.

Aldoss and colleagues retrospectively analyzed 65 patients with R/R ALL for predictors of leukemia response. The history of prior EM-ALL and active EM-ALL at the time of initiating blinatumomab predicted lower CR rates [20]. Among responders to blinatumomab, 20 (61%) subsequently relapsed among whom EM-ALL relapse occurred in 8 (40%) patients. Lau and colleagues also published their single-center experience with 20 adult patients with R/R-ALL. They observed 4 cases (20%) of EM relapse following blinatumomab [21]. On the other hand, a Korean group published two case reports of adults with isolated EM relapse after HSCT which were salvaged with blinatumomab monotherapy [22].

EM ALL might need to be treated with higher dosages of blinatumomab, in analogy to treatment of patients with non-Hodgkin lymphoma or combined with other agents such as inotuzumab ozogamicin or checkpoint inhibitors. Future trials will have to address this issue. Currently available data is much too scarce to draw a conclusion or recommend a therapeutic option.

Philadelphia-positive ALL

The European Medicines Agency currently restricts use of blinatumomab in children to Philadelphia chromosome negative (Ph-) B-cell precursor ALL, in adults therapy failure with at least two different tyrosine kinase inhibitors has to be documented.

For adult patients with Ph+ ALL there is convincing evidence for using blinatumomab in combination with tyrosine kinase inhibition. Foà and colleagues demonstrated that chemotherapy-free induction and consolidation for first-line treatment of Ph+ALL with dasatinib and blinatumomab is associated with high incidences of molecular response and survival and only few toxic effects (NCT02744768) [23]. The recently released long-term follow-up of blinatumomab in patients with R/R Ph+ ALL (ALCANTARA study NCT02000427) showed long-term durability of responses to blinatumomab [24].

Both recent RCT published this year for pediatric patients in first relapse did not include patients with Ph+ ALL. Colleagues from Memphis reported an interesting case report this year where they treated an adolescent with Philadelphia + ALL and cardiac disease (mechanical heart valves) with 5 cycles of blinatumomab and concomitant tyrosine kinase inhibition to limit treatment-related morbidity [25].

So far, the Australian data are the only to show relatively good outcomes for pediatric patients with Ph+ or Ph-like ALL, with a PFS of 70% [26].

As noted in our discussion section, there is unfortunately only one current front-line trial for children with ALL which includes Ph+ ALL (Total Therapy XVII, NCT03117751). We believe that this vulnerable subgroup of leukemia patients, suffering severe treatment-related morbidity should be eligible to receive blinatumomab in an upfront setting.